# Hypersensitized Metamaterials Based on a Corona-Shaped Resonator for Efficient Detection of Glucose

Yadgar I. Abdulkarim [1,2], Fahmi F. Muhammadsharif [3], Mehmet Bakır [4], Halgurd N. Awl [5], Muharrem Karaaslan [6], Lianwen Deng [1] and Shengxiang Huang [1,*]

1  School of Physics and Electronics, Central South University, Changsha 410083, China; Yadgar.kharkov@gmail.com (Y.I.A.); denglw@csu.edu.cn (L.D.)
2  Physics Department, College of Science, University of Sulaimani, Sulaimani 46001, Iraq
3  Department of Physics, Faculty of Science and Health, Koya University, Koya 44023, Iraq; fahmi.fariq@koyauniversity.org
4  Department of Computer Engineering Bozok University, 66200 Yozgat, Turkey; mehmet.bak@gmail.com
5  Department of Communication Engineering, Sulimani Polytechnic University, Sulaimani 46001, Iraq; halgurd.awl@spu.edu.iq
6  Department of Electrical and Electronics, Iskenderun Technical University, 31100 Hatay, Turkey; muharrem.karaaslan@iste.edu.tr
*  Correspondence: hsx351@csu.edu.cn

**Abstract:** In this work, a new design for a real-time noninvasive metamaterial sensor, based on a corona-shaped resonator, is proposed. The sensor was designed numerically and fabricated experimentally in order to be utilized for efficient detection of glucose in aqueous solutions such as water and blood. The sensor was inspired by a corona in-plane-shaped design with the presumption that its circular structure might produce a broader interaction of the electromagnetic waves with the glucose samples. A clear shift in the resonance frequency was observed for various glucose samples, which implies that the proposed sensor has a good sensitivity and can be easily utilized to distinguish any glucose concentration, even though their dielectric coefficients are close. Results showed a superior performance in terms of resonance frequency shift (1.51 GHz) and quality factor (246) compared to those reported in the literature. The transmission variation level $\Delta|S_{21}|$ was investigated for glucose concentration in both water and blood. The sensing mechanism was elaborated through the surface current, electric field and magnetic field distributions on the corona resonator. The proposed metamaterials sensor is considered to be a promising candidate for biosensor and medicine applications in human glycaemia monitoring.

**Keywords:** corona-shaped resonator; metamaterial sensor; dielectric characteristic; glucose concentration

## 1. Introduction

Glucose biosensors have been long used in biology, chemistry, food processing and diabetes diagnosis [1,2]. The developments of glucose biosensors, working on different principles, have been reported previously [3–10]. Among them, microwave-based biosensors have been widely used due to their high sensitivity, simultaneous measurement, fast response, robustness and low cost [11–14]. Glucose sensing by using microwave techniques is realized by means of detecting the resonance frequency shift or amplitude changes as a result of variation in the dielectric constant of the tested materials due to glucose content. This change in dielectric constant interacts with the time-varying electromagnetic field, which causes the resonance frequency to be shifted to a specific level [15–18].

Along this line, Lee et al. developed a reusable coplanar waveguide and found that the increase in glucose concentration led to a decrease of the penetration depth of the device [19]. Kumari et al. developed a resonator with complementary geometries of ring and horn shapes for glucose sample characterization [20]. Furthermore, Hassan et al.

showed that continuous detection of glucose was possible by using an (LC) inductor-capacitor tank resonator [21].

An open-ended spiral resonator was also presented [22]. This resonator is basically a spiral-shaped microstrip transmission line and has two ports. This study focused on a glucose tolerance test and tracked the transmission coefficient parameter (S21). Yilmaz et al. designed a patch resonator operating in the 2.45 GHz (ISM) industrial, scientific and medical band [23]. In that study, the input impedance of the resonator was tracked at the operating frequency to quantify the blood glucose change, which was approximately 0.04%. Consequently, three different frequencies were proposed [24] to test water and glucose solutions in the range from 0% to 10% and to track the change in glucose solutions by the Q factor. A ring resonator sensor proposed in [25] was used to calibrate the sensor for temperature changes in the sensing environment. An interface test system was proposed for both glucose and vitamins as well as other sugars present in the blood. Another resonator was designed by combining a spiral inductor and interdigital capacitor [26]. When blood plasma was placed on the resonator, the resonance frequency was shifted up if the glucose level was increased in the blood plasma. As such, a sensitivity of 199 MHz per mg/mL was reported in the study.

Some other designed antennas have also been proposed by scientists for glucose quantification. For instance, Freer and Venkataraman proposed an antenna through which reflection coefficient $S_{11}$ parameters are monitored in order to detect glucose in the operating frequencies of 1 GHz and 6 GHz [27]. Patch antennas operating at 2.45 GHz and 5.8 GHz were proposed in [28], in which deionized water and glucose solutions were analyzed. Results showed that the change was nonlinear and the antenna operating at a higher frequency was more responsive to the change in glucose levels. Another study proposed in [29] showed that using a pig blood digital phantom, the predicted shifts can range between 200 MHz to 300 MHz according to different volumes. A serpentine-shaped antenna with passive coupling was proposed in [30] to monitor the change in S11 response. This study showed that the designed antenna had a narrow bandwidth in air, while the bandwidth was increased to 2.6 GHz for simulation when a finger was used. When a glucose phantom was used, 32 MHz resonance frequency shifts were observed in the concentration range from 0 mg/dL to 200 mg/dL.

Since glucose content and the dielectric constant are directly correlated, a metamaterial sensor must satisfy the quality factor (Q) requirements for sensitivity reasons [31–33]. In order to increase the sensitivity, researchers usually consider different approaches to design the sensing resonators.

Based on the hypothesis that the symmetrical circular shape of a corona architecture might be helpful in generating a homogenous electromagnetic wave distribution across the resonator components, we expect that any trivial interaction of the external field with the dielectric behavior of the investigated sample would result in an interesting observation in the transmission response of the sensor. Therefore, in this study, we designed and fabricated a resonator based on a corona shape. The resonator dimensions were tuned by using a genetic algorithm embedded in the Computer Simulation Technology (CST) Microwave Studio program in order to optimize the whole design of the sensor. This work gained motivation from the fact that a highly sensitive glucose biosensor can be developed using a proper resonator design with a high Q factor [34]. It has been proved that when glucose concentration is changed, the effective permittivity ($\varepsilon_{eff}$) of the biosensor is ultimately modified, which in turn leads to the shift in resonance frequency or amplitude [35–41]. In the current work, the dielectric constant parameters were measured by an 85070E open-ended dielectric probe kit. The obtained parameters were simulated by the CST Microwave Studio program. The simulation and experimental studies showed good agreement. The findings of this study confirmed that glucose content in water or in blood can be efficiently sensed by the proposed metamaterial-inspired corona-shaped sensor.

## 2. Structure and Design of the Metamaterial-Based Sensor

The proposed metamaterial-based sensor was designed and numerically investigated by Computer Simulation Technology (CST) Studio Suite 2018. The designed structure used in this work is composed of three main layers, as shown in Figure 1. The top layer consists of a corona-shaped resonator with a transmission line, where P1 and P2 represent the first and second port, respectively. The space between the closed circular ring (CCR) and the conductive inner star shape is utilized as the sensor layer to be filled with samples under investigation. The top layer is made of copper with a conductivity of $5.8 \times 107$ S/m and thickness of 0.035 mm, while both ends of the transmission lines are excited by discrete port 1 and 2. The middle layer is a Rogers RT5870 substrate with a thickness of 1.6 mm and a dielectric constant and loss tangent of 2.33 and 0.0012, respectively. This substrate layer was chosen due to some advantages such as uniform electrical properties over a wide frequency range; lowest electrical loss; being able to be easily cut, shared and machined to the desired shape; being ideal for high-moisture environments; and being a well-established material. The bottom layer is fully covered with a copper metal of 0.035 mm thick, which behaves as a ground plate for the structure. The rest of the required dimensions are shown in Table 1.

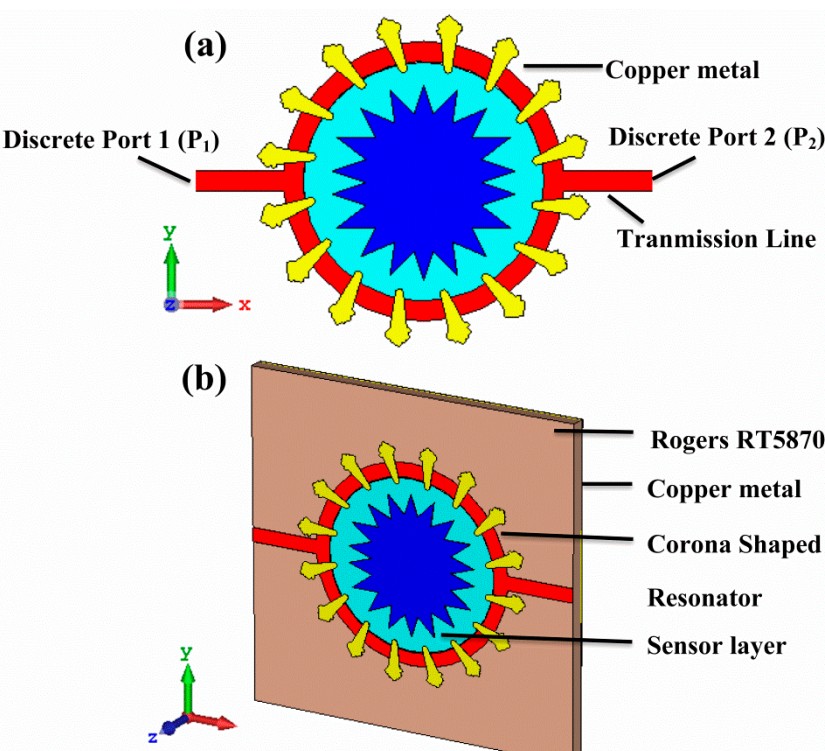

**Figure 1.** (**a**) Front view of the proposed metamaterials based on a corona shape and (**b**) its prospective view.

**Table 1.** Dimensions of the proposed structure.

| Dimensions | Size (mm) |
|---|---|
| Radius of resonator | 9 |
| Width of resonator | 1.5 |
| Length of transmission line | 7.5 |
| Width of transmission line | 1.5 |
| Radius of star shape | 7.3 |
| Length of substrate | 35 |
| With of substrate | 35 |

In the simulation software, there were some boundary conditions used depending on the desired applications, such as free space, periodic distribution, perfect electric conductor/perfect magnetic conductor (PEC/PMC) and perfect electric conductor (PEC). In this work, the boundary open (add space) was assigned at the X, Y and Z axes to be compatible with the experimental studies, as shown in Figure 2a.

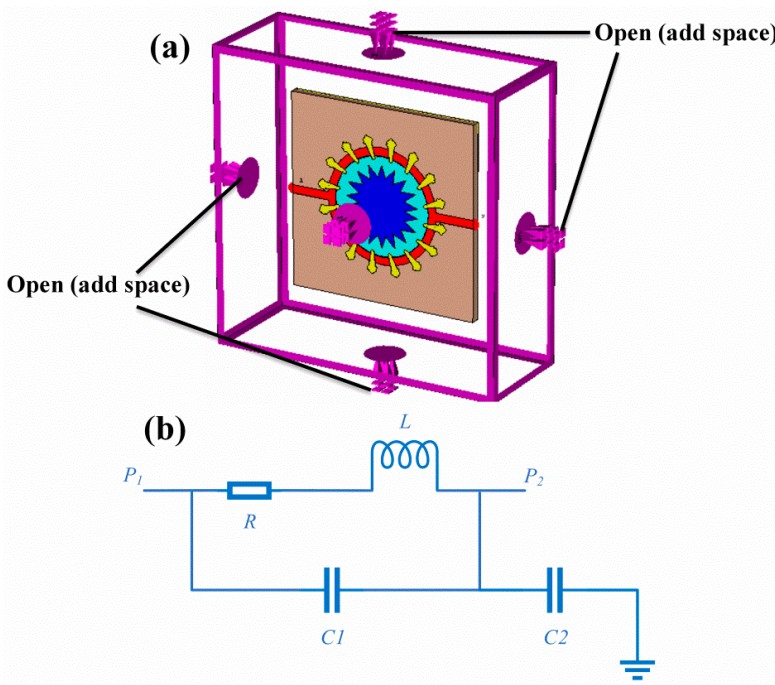

**Figure 2.** Metamaterial-based sensor with a corona-shaped resonator: (**a**) the boundary conditions and (**b**) equivalent circuit diagram whereas L denotes the inductance of the corona-shaped resonator, R is the resistance, and the capacitance between the outer ring and conductor is $C_1$, while $C_2$ is the capacitance between the inner conductor and copper ground plate.

To better understand the sensing mechanism of the proposed structure, an equivalent circuit diagram was drawn, as shown in Figure 2b. The designed structure has two ports, namely port 1 and port 2.

## 3. Dielectric Measurement of the Glucose–Water and Glucose–Blood Samples

The electrical properties of the samples were measured in the frequency range from 1 GHz to 8 GHz by using the open-ended coaxial probe with a 85070E dielectric measurement kit connecting to a vector network analyzer.

A Vector Network Analyzer (VNA) KEYSIGHT brand PNA-L N5234A and dielectric probe were used to measure the values of dielectric constant and loss tangent of the glucose mixture with water at different concentrations (100 mg/dL to 500 mg/dL) in steps of 100 mg/dL and of the glucose mixture with blood under similar conditions. The experimental setup, shown in Figure 3, was used for electrical measurements of the prepared samples. The measurements were carried out at room temperature (25 °C). Before starting the measurements of selected samples, the apparatus calibration in the VNA should be completed. To perform calibration, the electrical property of pure water was given to the VNA, while making sure that the dielectric measurement was idle and air was being measured. The next step was to immerse the dielectric probe kit in water and calibrate the device accordingly. Then, the electrical characteristic of water was measured in order to ensure the correct calibration of the VNA. Consequently, it was able to detect the real and imaginary parts of the relative permittivity for various mixtures of the selected samples. The dielectric loss factor of each sample could be determined by dividing the imaginary part of the dielectric constant ($\varepsilon''$) by its real part ($\varepsilon'$), $\tan \delta = \varepsilon''/\varepsilon'$.

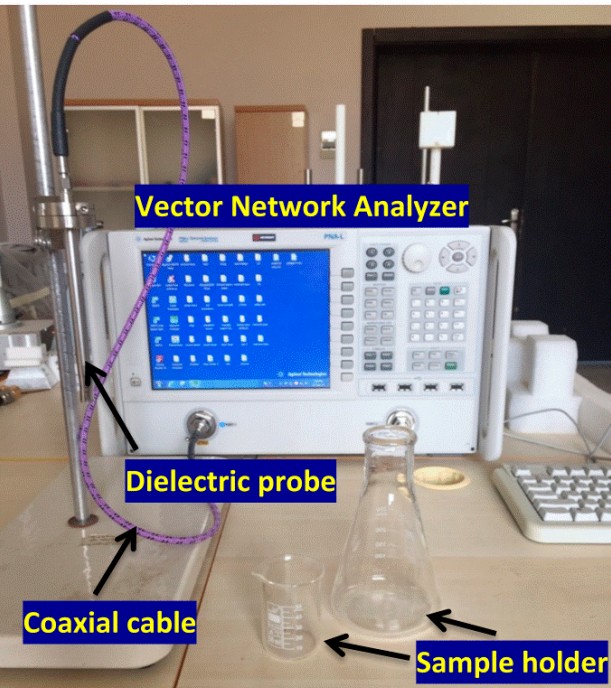

**Figure 3.** The experimental setup used in this work to measure the electrical properties of the samples.

The measured real part of the dielectric permittivity and the imaginary part of the glucose–water and glucose–blood mixtures were recorded and data were imported into the CST software to estimate the transmission coefficient. The measured real and imaginary parts of the complex permittivity for the glucose–water mixture are shown in Figure 4a,c. From the measured results, one can find the real part of the complex permittivity at any frequency from 1–8 GHz. The real part of the complex permittivity at a resonance frequency of 1 GHz for all glucose concentrations was estimated to be 80.5, while at the resonance frequency of 8 GHz, the real part of the complex permittivity for 100, 200, 300, 400 and 500 mg/dL glucose mixed with water was approximately 66.90, 67.91, 68.2, 69 and 69.92, respectively.

To verify the measured results, the electromagnetic parameters, namely the infinite dielectric constant ($\varepsilon_\infty$), static dielectric constant ($\varepsilon_s$) and relaxation time ($\tau$), were theoretically deduced from Debye's model. The formulas for the parameter's correlation with the concentration ratio of glucose were found to be as follows:

$$\varepsilon_\infty(x) = -0.9983x^2 - 0.3278x + 41.077 \tag{1}$$

$$\varepsilon_s(x) = -1.9436x - 1.2992x + 1.4725 \tag{2}$$

$$\tau(x) = 3E - 13x^2 - 3E - 12x + 1E - 11 \tag{3}$$

where x is the volume fraction of the glucose concentration in water. The calculated parameters can be inserted into Debye's equation so as to yield a generalized formula which can be used to determine the real and imaginary permittivity of the samples at various volume fractions, as follows:

$$\varepsilon'(w) = \varepsilon_\infty(x) + \frac{\varepsilon_s(x) - \varepsilon_\infty(x)}{1 + jw\tau(x)} \tag{4}$$

$$\varepsilon''(w) = \frac{[\varepsilon_s(x) - \varepsilon_\infty(x)]w\tau}{1 + jw\tau(x)} + \frac{\sigma_s}{w\varepsilon_o} \tag{5}$$

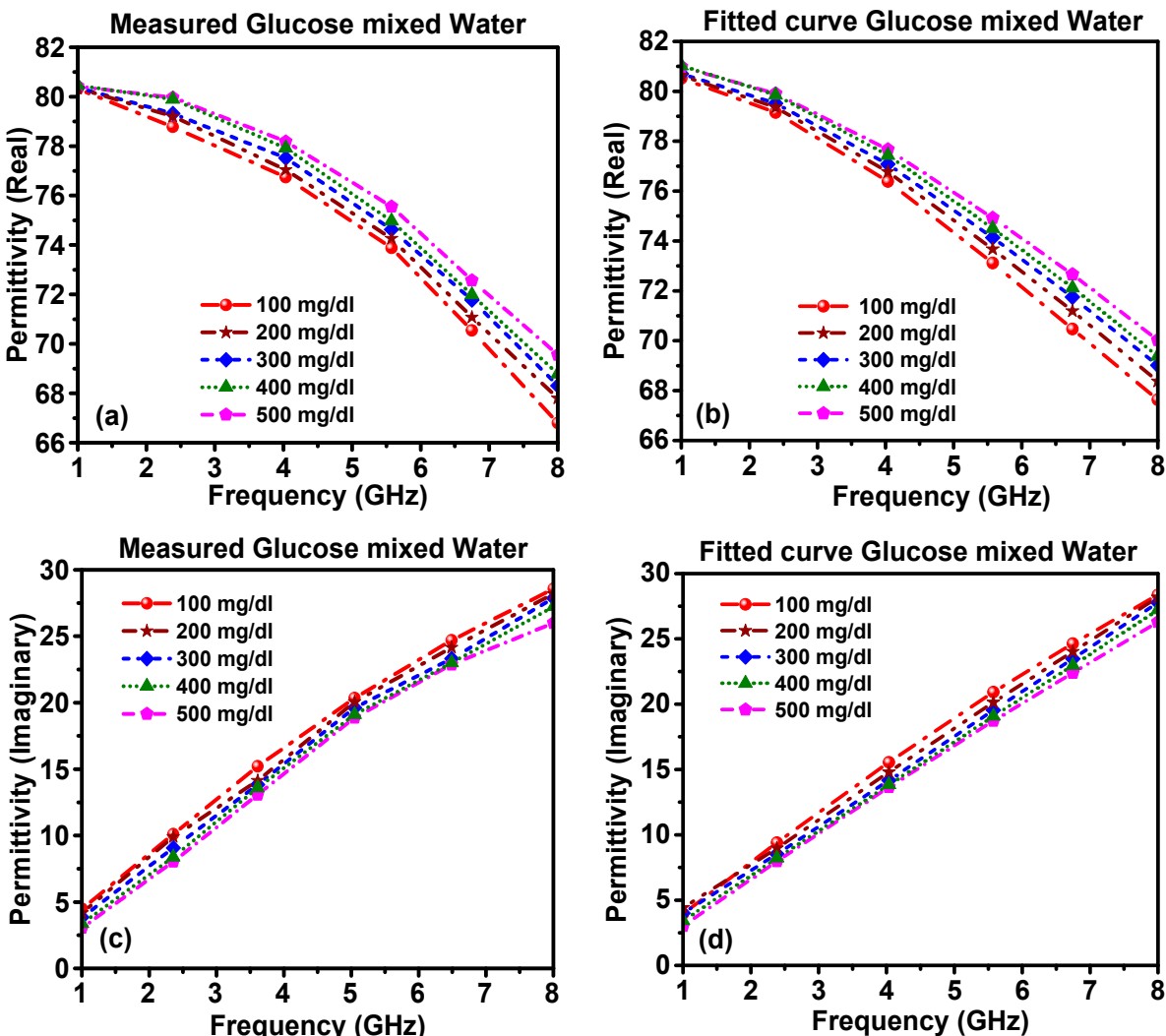

**Figure 4.** The complex permittivity of glucose–water mixture: (**a**) measured real part, (**b**) calculated real part, (**c**) measured imaginary part and (**d**) calculated imaginary part.

The theoretically extracted real part and imaginary part of the complex permittivity values are shown in Figure 4. The real part of the complex permittivity for the samples with 100, 200, 300, 400 and 500 mg/dL of glucose mixed with water was 80.5, 80.5, 80.5, 81 and 81, respectively, at the resonant frequency of 1 GHz. At the resonance frequency of 8 GHz, the values were 67.8, 68.2, 68.96, 69.53 and 70 for the same concentrations, respectively. The real and imaginary parts of the complex permittivity for different ratios of glucose–water showed a good agreement between the measured values and theoretically extracted ones from Debye's equation.

To further validate the measured and theoretical results, we conducted an analysis of the real and imaginary parts of the complex permittivity of the glucose–blood samples, as shown in Figure 5. The measured and calculated results were in good agreement. The calculated curves of the real and imaginary permittivity versus frequency in the range of 1–8 GHz were derived theoretically from Debye's equation, Equations (4) and (5), after estimating the following parameters, Equations (6)–(8), upon the curve fitting of the experimental data.

$$\varepsilon_\infty(x) = -66.171x^3 + 124.98x^2 - 65.129x + 47.224 \tag{6}$$

$$\varepsilon_s(x) = -136.39x^3 + 225.95x^2 - 130.14x + 20.669 \tag{7}$$

$$\tau(x) = 2E - 12x^2 - x \times 10^{-13} + 10^{-11} \tag{8}$$

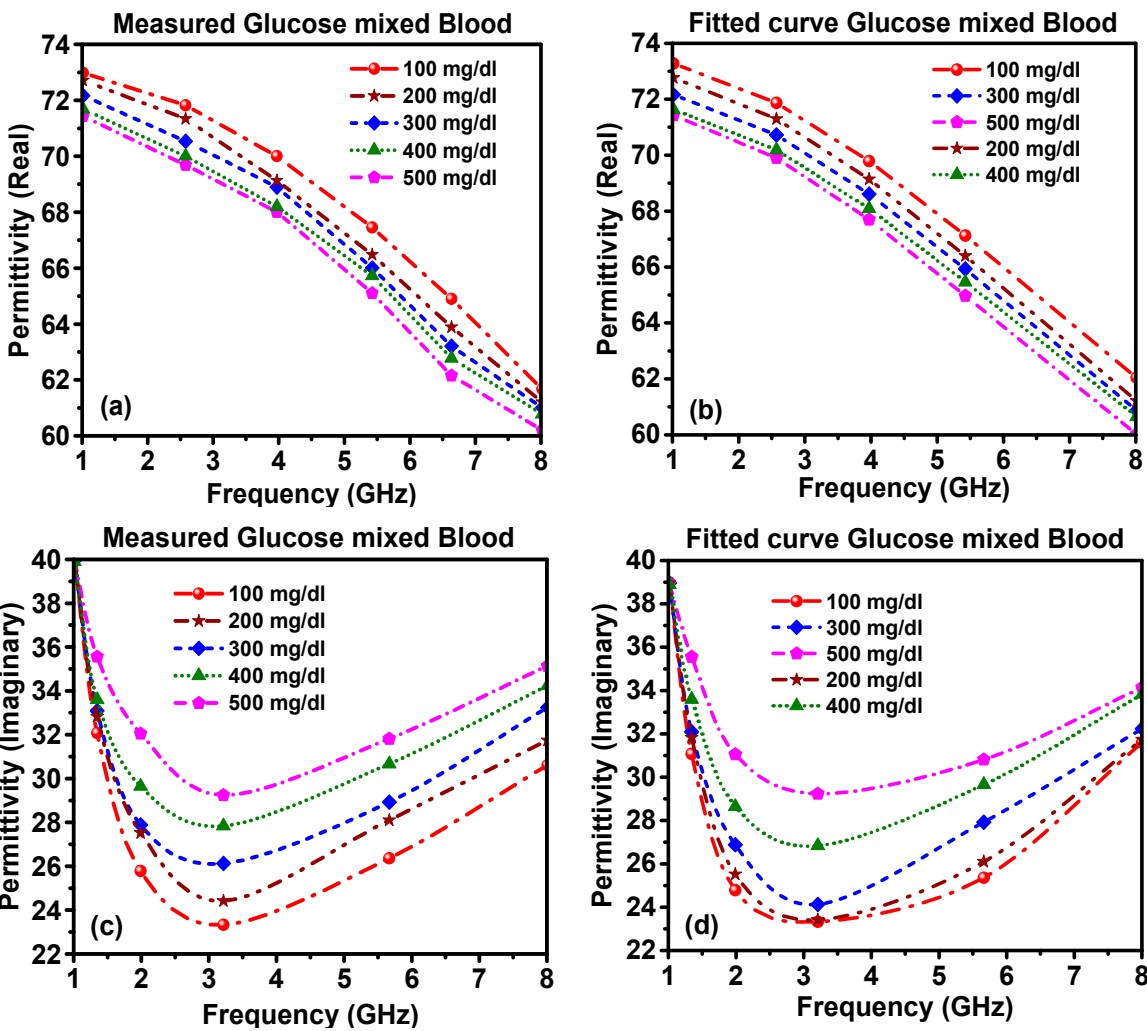

**Figure 5.** The complex permittivity of glucose–blood mixture: (**a**) measured real part, (**b**) calculated real part, (**c**) measured imaginary part and (**d**) calculated imaginary part.

## 4. Numerical and Experimental Studies of the Proposed Sensor for Glucose Detection

The designed (MTM) metamaterials corona-shaped sensor was simulated by the CST Studio suite electromagnetic solver. When the sensing layer was filled with air, the reflection coefficient ($S_{11}$) and transmission coefficient ($S_{21}$) were monitored over the frequency range of 1–8 GHz, as shown in Figure 6. It can be noted from Figure 6a that the $S_{11}$ had two resonance frequencies at 3.6 GHz and 6.2 GHz with −32 dB and −26 dB, respectively. These correspond to the two maximal values of the transmission coefficient at the same frequencies in Figure 6b. These two sharp peaks of the transmission and reflection coefficient are considered to be one of the important advantages of the proposed design, which helps to detect any trivial shift/change in the resonance frequency.

In order to optimize the performance of the proposed sensor, a parametric study was performed to tune the sensor dimensions. For this purpose, the overall dimensions of the proposed structure were optimized by using a genetic algorithm technique, which is a built-in function in the CST software. To address the effects of changing the dimensions of the proposed sensor, the most influential parameters, which are the width of the outer ring resonator radius (Ro), transmission line width (WTI) and sensor layer radius (Rs), were investigated, as shown in Figures 7–9.

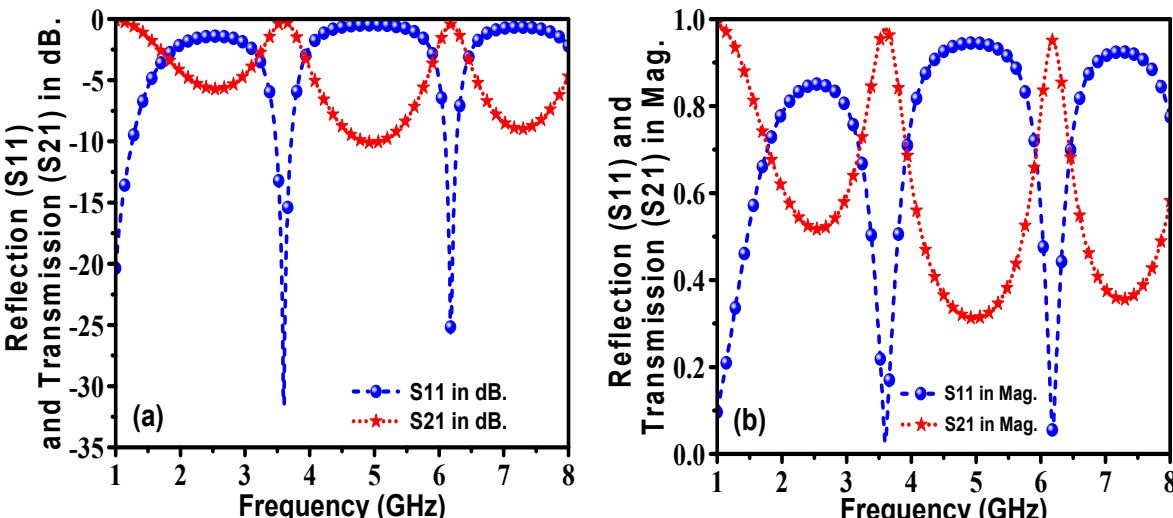

**Figure 6.** Simulated results of the reflection and transmission coefficients for the proposed corona-shaped sensor when the sensor layer was filled with air: (**a**) in dB and (**b**) in magnitude.

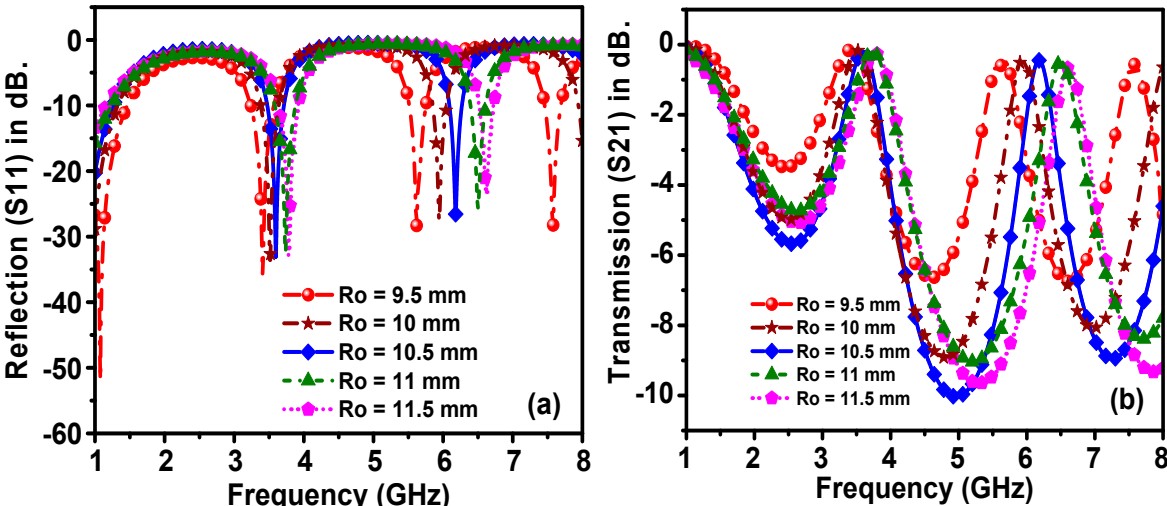

**Figure 7.** Effect of the width of the outer ring resonator radius on the resonant frequency of (**a**) reflection and (**b**) transmission spectra.

From the studied results, a clear shift in the resonance frequencies of the reflection coefficient to the upper-frequency limit was noted when the value of Ro was increased from 9.5 mm to 11.5 mm, as shown in Figure 7. Besides, the Ro increment led to the amplitude variation and increment of the resonance frequency of the transmission coefficient.

The effects of changing the transmission line width (WTI) from 0.5 to 2.5 mm are presented in Figure 8. As can be seen from the figure, with WTI decrement in 0.5 steps, the resonance frequency of the reflection coefficient increased by a small amount. Noticeably, the amplitude of the transmission coefficient was reduced with this WTI reduction, while this had a very small effect on the resonance frequency shift of the transmission coefficient.

Figure 9 shows the variation of the resonance frequency and amplitude of the reflection coefficient and transmission coefficient of the proposed sensor when the sensor layer radius was changed from 8.4 to 9.6 mm in 0.3 mm steps. The result showed that there is a direct proportionality relation between the sensor layer radius and the resonance frequency of the reflection coefficient. When the radius is increased, the resonance frequency is also increased linearly by about 100 MHz in each step. Consequently, based on the optimization of the genetic algorithm technique, the optimum values of the outer ring resonator radius

($R_o$), transmission line width (WTI) and sensor layer radius ($R_s$) were found to be 10.5 mm, 1.5 mm and 9 mm, respectively.

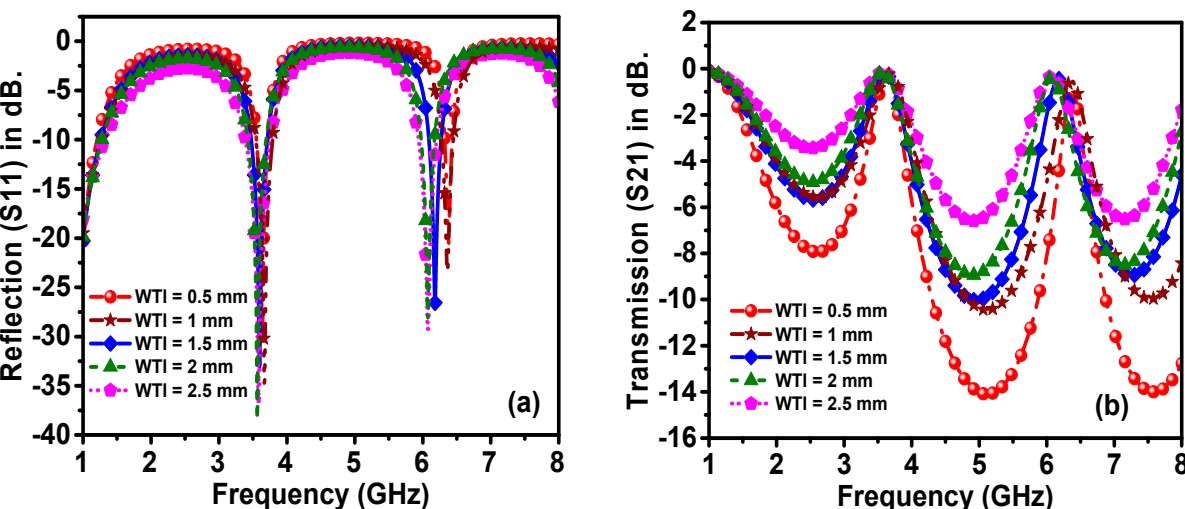

**Figure 8.** Effect of transmission line width on the resonance frequency of (**a**) reflection and (**b**) transmission spectra.

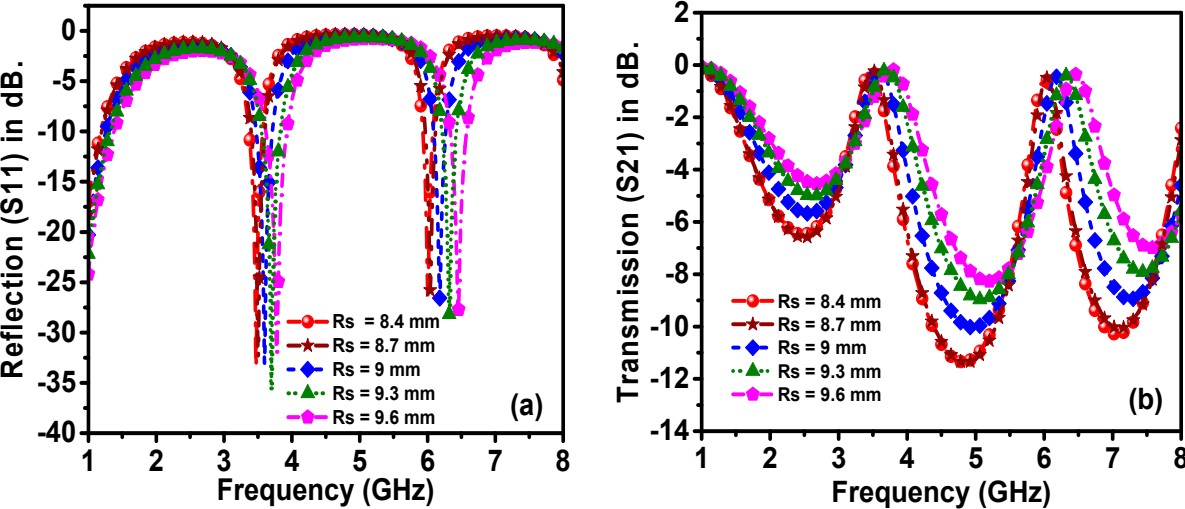

**Figure 9.** Effect of the sensor layer radius on the resonant frequency of (**a**) reflection and (**b**) transmission spectra.

In conclusion, it can be said that the positions of the resonance frequencies are mostly affected by the outer ring resonator radius ($R_o$) and the sensor layer radius. This implies that these two parameters are the most influential parameters of the proposed structure, as they are highly related to the investigated samples. Consequently, any minor change in the electrical characteristics of the samples led to the shift in resonance frequency.

The measurements were run seven times sequentially and the average of the measurements was taken. Hence, the standard deviation of the measurements is represented by error bars on the corresponding plots. The proposed designed structure is shown in Figure 10a. The required dimensions of the structure were fabricated based on the optimized dimension achieved in the numerical design. Additionally, the substrate length and width were chosen to be 35 mm × 35 mm. Two coaxial test cables were connected to the metamaterial sensor via two ports, as shown in Figure 9b. Before taking measurements, the vector network analyzer (VNA) was connected to the structure through the coaxial cables and then calibration was done in three steps of the open circuit, short circuit and 50 Ohm load connector in the desired frequency range, as shown in Figure 9c. To take the reference data, a simulation and experiment were carried out with the presence of the glucose–water

and glucose–blood samples in the sensor layer. The test results were done in the frequency range of 1–8 GHz.

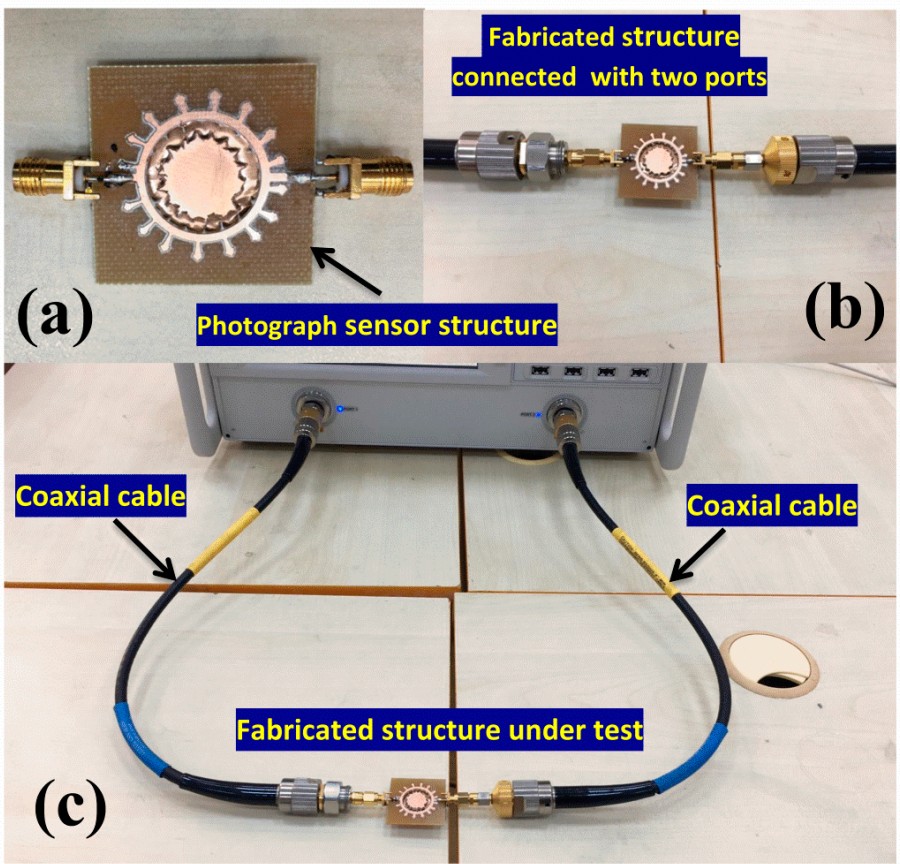

**Figure 10.** Photographs of (**a**) the proposed structure, (**b**) its two-port connection and (**c**) experimental setup.

Glucose–water mixtures were prepared at different concentrations of glucose, from 100 mg/dL to 500 mg/dL in steps of 100 mg/dL. The dielectric properties of the samples were measured using a dielectric probe kit. The data files obtained from the measured results were imported into simulation software to obtain the transmission coefficient ($S_{21}$) in dB, in the frequency range of 1–8 GHz. The sensitivity of the proposed sensor was examined by means of the transmission coefficient value for all glucose concentrations, and the simulated results are illustrated in Figure 11a. As can be seen from the figure, when the glucose concentration in water was about 100 mg/dL, the resonance frequency shift was about 7.01 GHz. However, by increasing the concentration of the glucose, a clear shift in the resonance frequency could be seen towards the lower frequency. For instance, at 500 mg/dL, the resonance frequency was about 6.25 GHz, while in the frequency range from 1 GHZ to 1.6 GHz, a shift in the resonance frequency was observed when the glucose concentration was increased from 100 mg/dL to 500 mg/dL. The inset of Figure 11a shows the close-up for this shift. To validate the numerical results of the proposed sensor, the measured results of the transmission coefficient ($S_{21}$) for the glucose–water mixtures at similar concentrations was recorded, as depicted in Figure 11b. The measured results showed that the designed structure was very sensitive to the samples inside the sensor layer; as seen from the figure, there was a clearer shift for all the concentrations. For instance, the resonance frequency shift between 100 mg/dL and 200 mg/dL was about 1.51 GHz. This frequency shift is larger than the shifts achieved in the other similar published works in the literature. In terms of the Q-factor, the proposed structure had a superior quality factor, which was about 246 based on the measured results at the glucose concentration of 300 mg/dL in water.

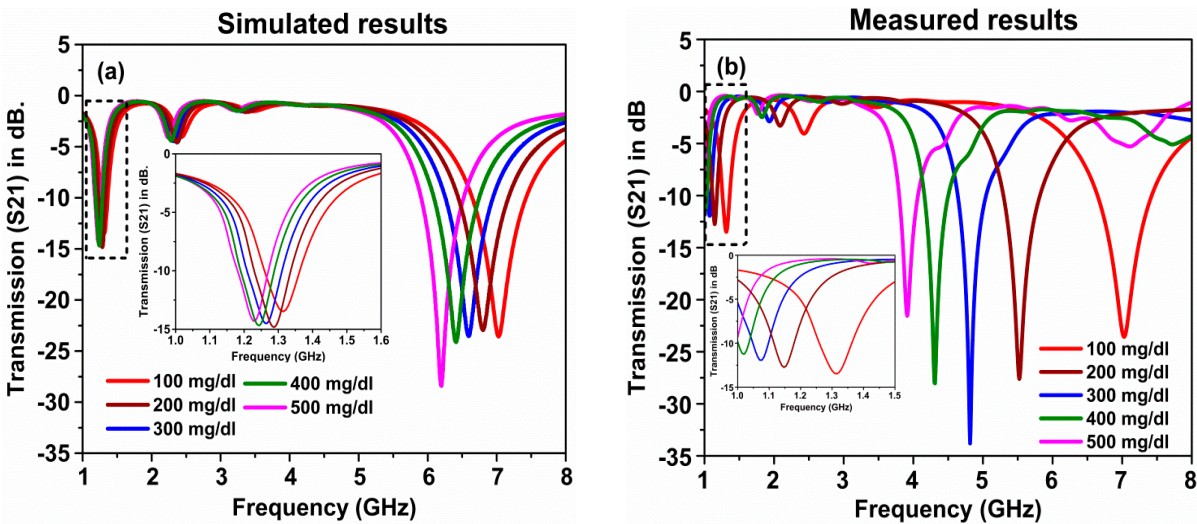

**Figure 11.** (**a**) Simulated and (**b**) measured results of the glucose–water mixture of the proposed metamaterials sensor.

In order to verify the sensitivity of the proposed sensor, the simulation and measurements were performed with other glucose concentrations mixed with blood: 100, 200, 300, 400 and 500 mg/dL as illustrated in Figure 12. The simulated and measured results were found to be in good agreement, where there were clear shifts in the resonance frequency of both numerical and measurement results around 2 GHz and 3.5 GHz. The results had deeper peaks around $-16$ dB. Both results indicated that the proposed corona-shaped sensor could easily detect the minimal glucose concentration in blood, which is very important for the medical and biological applications.

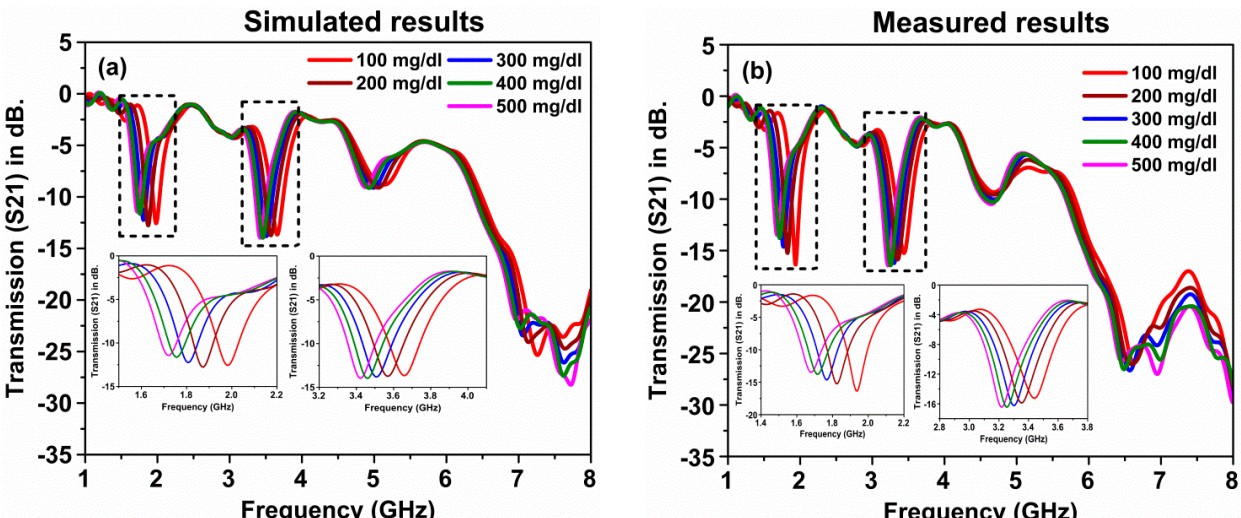

**Figure 12.** A comparison of the glucose–blood mixture sample of the proposed metamaterials sensor: (**a**) simulated and (**b**) measured.

Figure 13a shows the comparison of the simulated and measured results along with the fitted curve of the resonance frequency shift versus glucose concentration in aqueous solution. The resonance frequency of the proposed sensor shifted down by increasing the concentration of glucose in water; the frequency shift also caused a change in the transmission coefficient ($S_{21}$) level at a fixed frequency of 7.01 GHz, which is the resonance frequency for the 100 mg/dL glucose–water mixture. Therefore, both the resonance frequency shift ($\Delta f_r$) and the $S_{21}$ level can be used in the sensing mechanism. The resonance frequency shift versus the glucose concentration is plotted in Figure 13a. The resonance frequency shifts of the experimental results and fitted curve are plotted in Figure 13b. It

was noticed that by increasing the glucose concentration in water, the resonance frequency shift was decreased. The lowest concentration of glucose which was reliably detected by the proposed sensor can be expressed by the limit of detection (LOD). The value of the LOD defines the lowest detectable quantity of an analyte when the concentration approaches to zero. The response of the sensor in the range of 200–500 mg/dL was found to be well fitted as y = 6.414 − 0.0048x ($R^2$ = 0.972). Consequently, the LOD was determined to be 30.94 mg/dL using LOD = $K \times S_0/S$ [42], where $K$ was chosen as 3.3 with respect to the 95% confidence level, $S_0$ is the standard deviation and $S$ is the slope of the fitting curve.

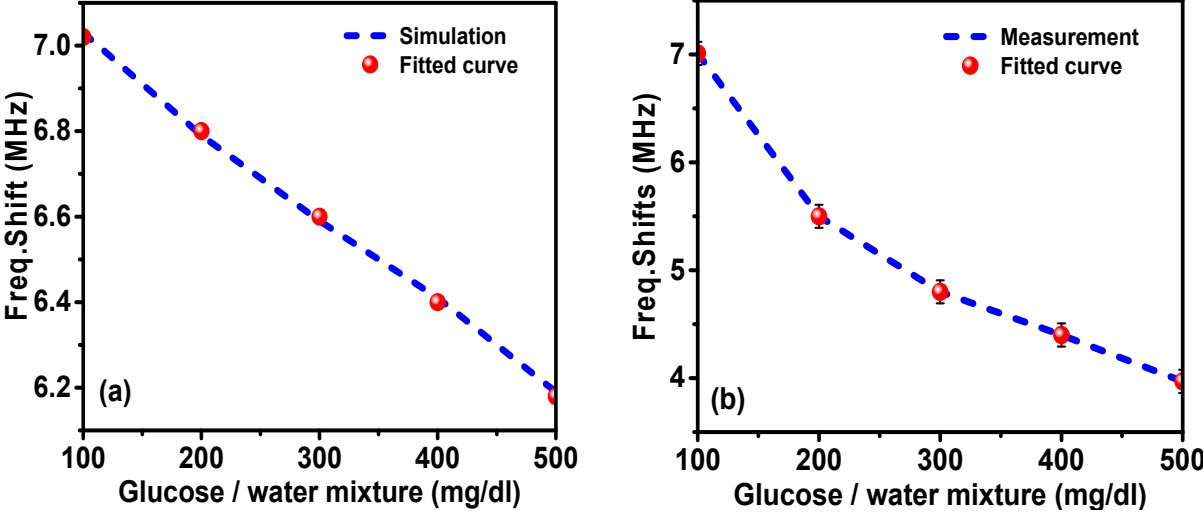

**Figure 13.** Resonance frequency shift versus the glucose concentration in water: (**a**) simulation and fitted curve, and (**b**) measurement and fitted curve.

Figure 14 shows the frequency shift dependence of the glucose–blood mixtures for the simulated and measured results. It was seen that the fitted curve was in compliance with the resonance frequency shifts, concluding that the resonance frequency shifts are decreased by increasing the glucose concentration in blood.

Figure 15a shows the change in the transmission level variation, $\Delta|S_{21}|$, with respect to the 100 mg/dL glucose–water mixture at 7.01 GHz for the simulation and fitted curve derived from the Debye formula. The results show a linear relationship between the resonance frequency shift and the glucose concentration in the solution. However, the $S_{21}$ level variation is a nonlinear function of the glucose concentration for the considered concentration ranges. The points in Figures 13a, 14a, 15a and 16a were obtained by repeating each simulation 10 times. As it can be seen from the figure, the variation in transmission level was increased when the glucose concentration in water was increased. To verify the simulated results, the experimental results were recorded and compared with the numerical results. The measurement and fitted curve of the transmission variation level with respect to the 100 mg/dL mixture as a reference at 7.01 for the glucose–water mixture was obtained and plotted in Figure 15a. The results show a linear relationship between the transmission variation level and glucose concentration in water.

The simulated and measured results of the variation in transmission level with respect to the glucose concentration were recorded for the samples of glucose–blood, as shown in Figure 16a. The results showed a linear relationship between the transmission level and glucose concentration in blood.

Figure 16b shows the dependence of the transmission variation level on glucose concentration in blood for the measured results and fitted curve. The glucose concentration of 100 mg/dL at resonance frequencies of 2 and 3.5 GHz was used as a reference. The measured and fitted curve were in good agreement, indicating the linear relationship between the transmission variation level and glucose concentration.

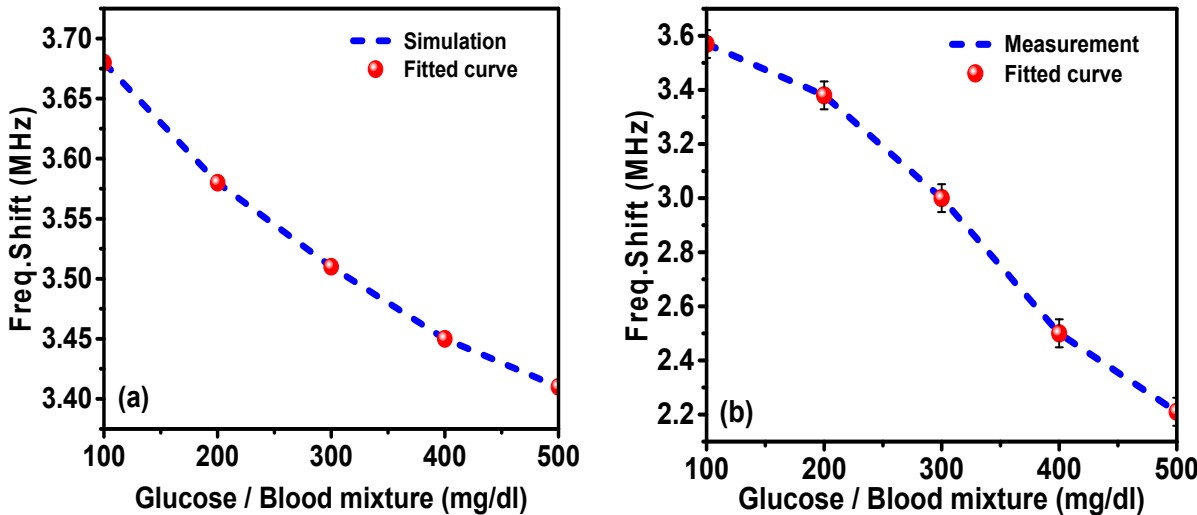

**Figure 14.** A comparison of resonance frequency shift versus the glucose concentration in blood: (**a**) simulation and (**b**) measurement.

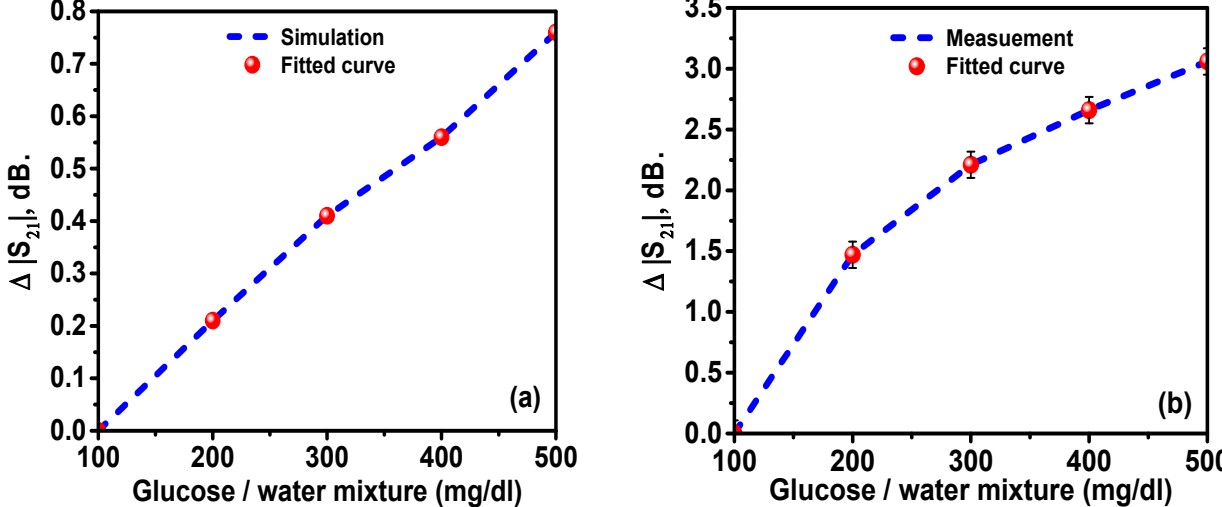

**Figure 15.** Transmission level variation, $\Delta|S_{21}|$, in dB as a function of the glucose concentration in water: (**a**) simulated and (**b**) measured.

To better understand the working principle of the proposed metamaterials sensor, the simulated surface current, electric field and magnetic field distribution was studied at two different resonance frequencies of 3.6 and 6.18 GHz, as shown in Figures 17–19, respectively. At the resonance frequency of 3.6 GHz, the current flowed into port one and exited from port two. Furthermore, at the lower part of the resonator, the direction of the current flow was clockwise, while at the upper part of the resonator, the current flow was in an anticlockwise direction, as shown in Figure 17a. At the resonance frequency of 6.18 GHz, the current was heavily distributed on the resonator, whereas currents on port one and two were flowing in opposite directions and both the resonator and transmission line were in parallel and anti-parallel directions. Nevertheless, the parallel currents control the electric response and the anti-parallel currents control the magnetic response, as shown in Figure 17b.

Figure 18 depicts the simulated electric field distribution for the proposed metamaterials sensor at two different resonance frequencies of 3.6 and 6.18 GHz. From Figure 18a, it is apparent that the electric field intensity is mostly concentrated in the inner and outer rings of the resonator. In addition, it was localized at the transmitting side (port 1) of the transmission line compared with at the other side at the resonance frequency of 6.18 GHz,

as shown in Figure 18b. Electric field distribution was highly concentrated on the resonator (especially the inner and outer rings). Hence, the proposed structure was able to sense any small changes in the electrical characteristics of the sample placed in the sensor layer.

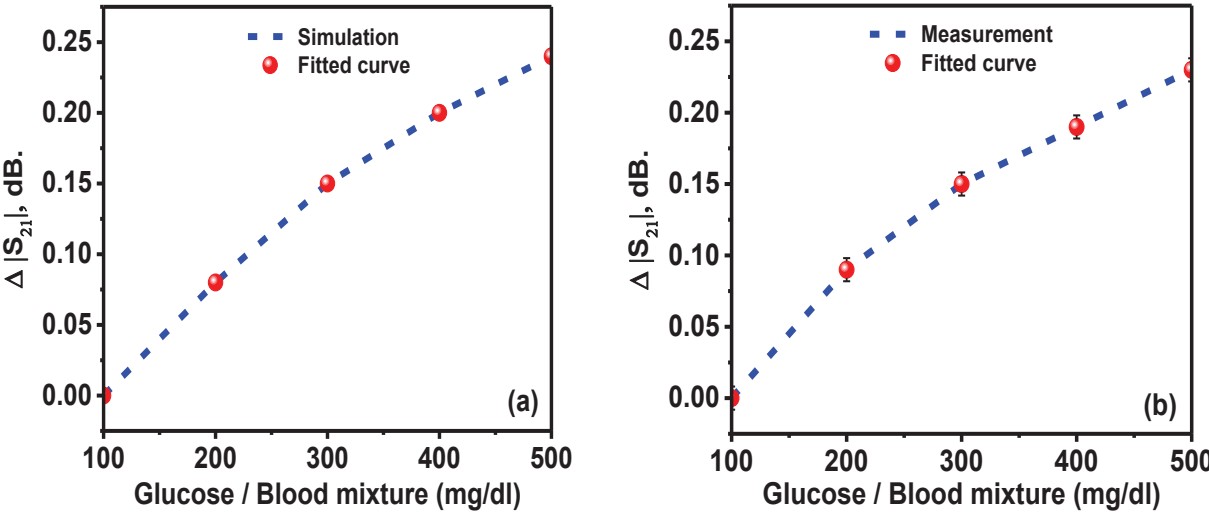

**Figure 16.** (**a**) Simulated and (**b**) measured transmission level variation, $\Delta |S_{21}|$, dB against the glucose concentration in blood.

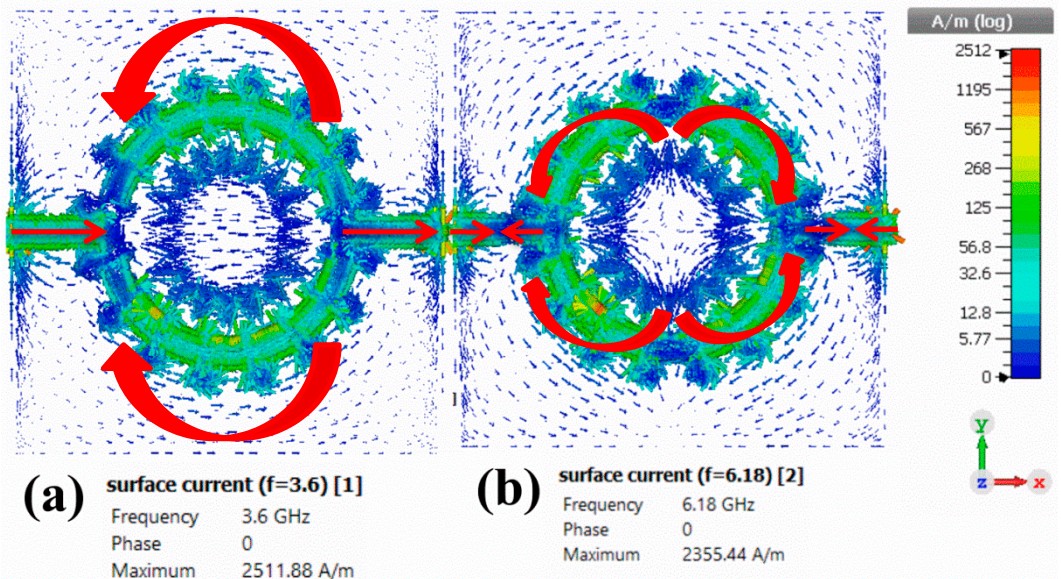

**Figure 17.** Simulated surface current distribution for the proposed corona-shaped sensor at (**a**) 3.6 GHz and (**b**) 6.16 GHz.

The simulated magnetic field distribution for the proposed sensor at resonance frequencies of 3.6 GHz and 6.18 GHz is shown in Figure 19. The magnetic field distribution was strongly localized at the upper and lower parts of the resonator at the resonance frequency of 3.6 GHz, as shown in Figure 19a. However, when the resonance frequency was increased to 6.18 GHz, the magnetic fields were mostly concentrated at the resonator as shown in Figure 19b.

A comparison between this work and other similar published work in term of size, materials substrate, frequency range, sensitivity and application procedure has been shown in Table 2.

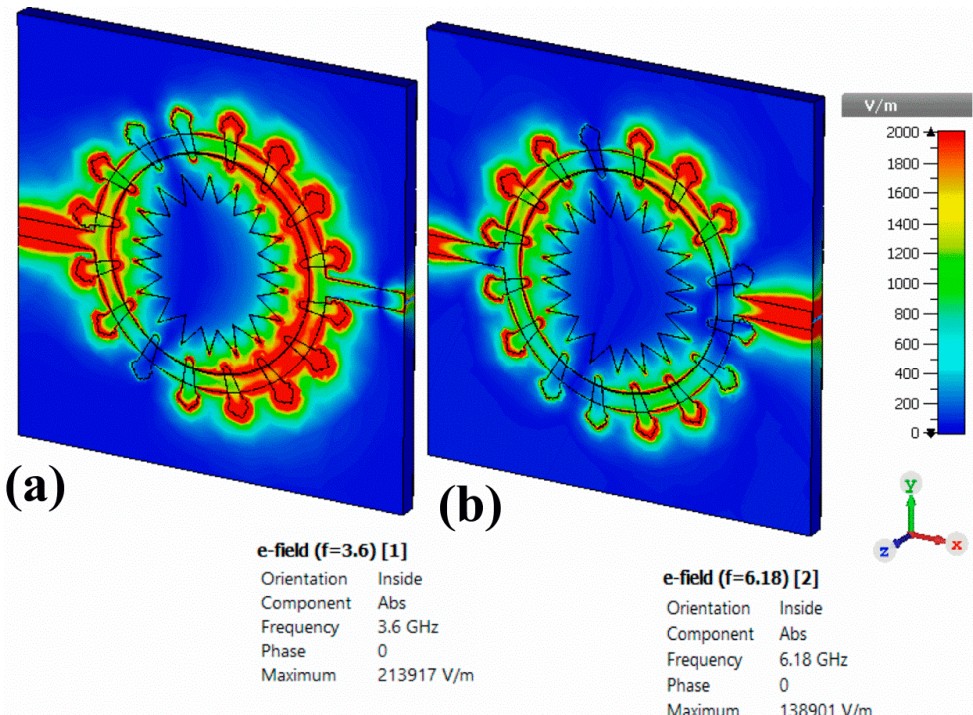

**Figure 18.** Electric field distribution for the proposed corona-shaped sensor at two different resonance frequencies: (**a**) 3.6 GHz and (**b**) 6.16 GHz.

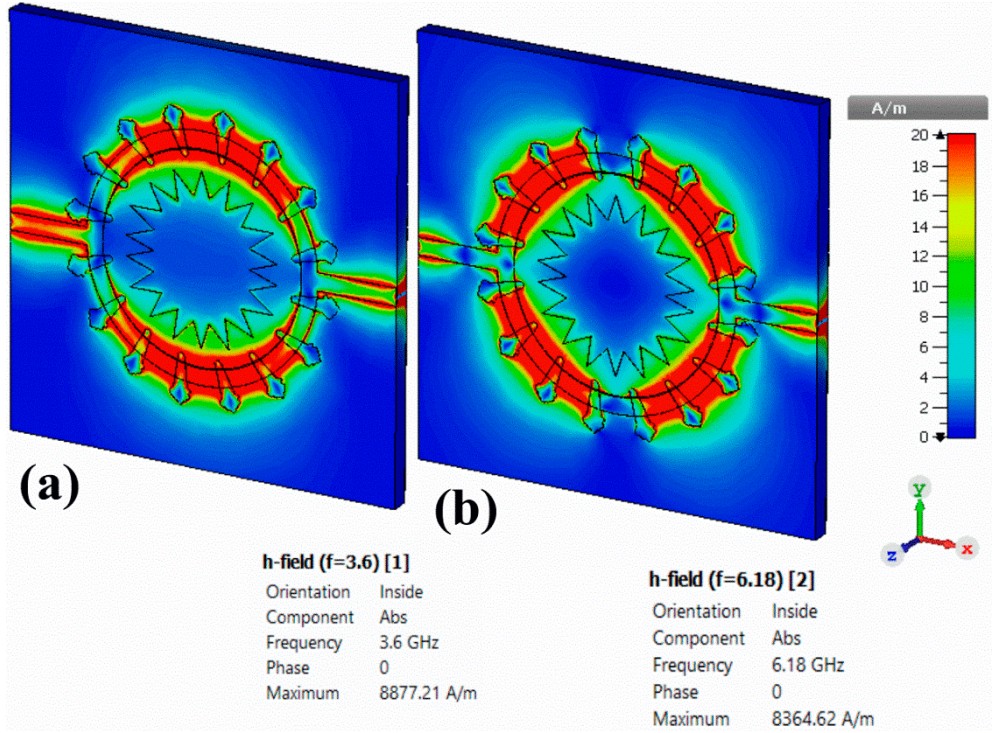

**Figure 19.** Simulated magnetic field distribution for the proposed corona-shaped sensor at (**a**) 3.6 GHz and (**b**) 6.16 GHz.

**Table 2.** Comparison study of the proposed glucose sensor with those reported in literature.

| Ref. | Size (mm) | Substrate Materials | Frequency Range (GHz) | Sensitivity | Application Procedure | Remarks |
|:---:|:---:|:---:|:---:|:---:|:---:|:---|
| [6] | 46 × 46 | FR-4 | 0.7–1.2 | 0.033 MHz g/L | Noninvasive | Split Ring Resonator (SRR)-based microwave fluidic sensors |
| [7] | 20 × 20 | Rogers RO4350B | 2–5 | 0.037 GHz 30 mg/dL | Noninvasive | Sensing capacity with double negative (DNG) property and minimal absorption |
| [10] | 40 × 40 | Silicon | 50–67 | Range 2.2–7.7 mg/mL | Noninvasive | Whispering Gallery Modes (WGMs) launched in a dielectric disc resonator (DDR) |
| [17] | 50 × 20 | Rogers RT6006 | 1–5 | 0.026 MHz mg/dL | Noninvasive | Metamaterial-inspired microwave microfluidic SRR |
| [18] | 20 × 15 | Rogers RT5880 | 1–2 | 1.6 MHz1–15 g/dL | Glucose-sensing | SRR resonator without metamaterials |
| [20] | 40 × 20 | FR-4 | 1–3 | Range 20–100 mg/mL | Invasive | Microwave filter as a sensor device |
| This work | 35 × 35 | Rogers RT5880 | 1–8 | 1.51 GHz 100–500 mg/dL | Glucose-sensing | Corona resonator based on metamaterials |

## 5. Conclusions

In this work, for the first time, a metamaterial-based sensor comprising a corona-shaped resonator was successfully developed for the efficient detection of glucose concentration. The sensor was designed numerically and tested experimentally by evaluating variations in the transmission coefficient ($S_{21}$) of the waves at resonant frequency. The proposed structure can offer a noninvasive characterization technique based on the dielectric properties of the measured glucose concentration in water or blood. According to the measured results for the glucose concentration in water, the resonance frequency shift for concentrations between 100 mg/dL and 200 mg/dL was about 1.51 GHz. A favorable quality factor of 246 could be achieved, which is highly competitive compared to that of previous results. The proposed metamaterial-based sensor has many advantages, such as its use in real time, low cost, durability, accuracy and ability to detect any glucose concentration in the samples in a few seconds. The proposed sensor can be considered for many applications of biosensing and medicine and in monitoring human glycaemia.

**Author Contributions:** Y.I.A. and F.F.M. conceived the idea, Y.I.A. and H.N.A. performed the simulations, and M.B. performed the experiments. Y.I.A. wrote the manuscript; L.D., F.F.M. and M.K. revised the manuscript; and S.H. supervised this research. All authors have read and agreed to the published version of the manuscript.

**Funding:** This research was funding by the National Key Research and Development Program of China (Grant No. 2017YFA0204600), the National Natural Science Foundation of China (Grant No. 51802352), Central South University (Grant No. 2018zzts355) and Teaching reform for postgraduate students of Central South University (Grant No. 2019JG085).

**Institutional Review Board Statement:** The study was conducted according to the guidelines of the Declaration of Helsinki, and approved by the Institutional Review Board of Iskenderun Technical University.

**Informed Consent Statement:** Informed consent was obtained from all subjects involved in the study.

**Data Availability Statement:** No new data were created or analyzed in this study. Data sharing is not applicable to this article.

**Conflicts of Interest:** The authors declared that they have no conflict of interest to this work.

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
