# Peer review of "Hypersensitized Metamaterials Based on a Corona-Shaped Resonator for Efficient Detection of Glucose"

_applsci, doi:10.3390/app11010103_

Round 1

Reviewer 1 Report

The researchers designed and fabricated a glucose detecting sensor based on coronavirus-shaped resonator. This work is scientific and interesting. The demonstrated results are significant and repeatable. Although the major materials used in this study are commercially devices, the works are sufficient enough for a publication. Suggest to accept after a minor revision.

  • Error bars of raw data should be added in relevant figures.
  • Figure 3 & 10, some annotation should be added in the picture for a better readability.
  • Coronavirus-shape is a negative word. Find a subsitute for describing this shape.

Author Response

Reviewer 1

Comments and Suggestions for Authors

The researchers designed and fabricated a glucose detecting sensor based on coronavirus-shaped resonator. This work is scientific and interesting. The demonstrated results are significant and repeatable. Although the major materials used in this study are commercially devices, the works are sufficient enough for a publication. Suggest to accept after a minor revision.

(1) Error bars of raw data should be added in relevant figures.

Response:

Thank you for the comment. The error bar has been added to the measured results as shown in Figures 13(b), 14(b), 15(b), 16(b).

(2) Figure 3 & 10, some annotation should be added in the picture for a better readability.

Response:

Thank you for the comment. Necessary annotations were added to Figure 3 & 10 in the revised manuscript.

 (3) Coronavirus-shape is a negative word. Find a substitute for describing this shape.

Response:

Thank you for your suggestion. We have removed the virus word and the coronavirus-shaped term was replaced by corona-shaped one throughout the manuscript.

Reviewer 2 Report

Authors do an thorough job describing their development of a glucose sensor. This work will be of use to many in the medical fields. There are minor issues with their grammar that need to be addressed.

Authors need to mention how many times banch-top experiments were performed and include error bars in their plotted data, and standard deviations in their tables.

Authors should show both a minimal (0mg/dL glucose) and maximal range of their sensor, to better highlight the efficacy of this system to medical professionals and as a sign of the sensitivity of their machine

Author Response

Reviewer 2

Comments and Suggestions for Authors

Authors do an thorough job describing their development of a glucose sensor. This work will be of use to many in the medical fields. There are minor issues with their grammar that need to be addressed.

(1) Authors need to mention how many times banch-top experiments were performed and include error bars in their plotted data, and standard deviations in their tables.

Response:

Thank you for the comment. The measurements were run seven times sequentially and the average of the measurements was taken. Hence, the standard deviation of the measurements was represented by error bars on corresponding plots. The error bars have been added to the Figures 13(b), 14(b), 15(b), 16(b) accordingly.

 (2) Authors should show both a minimal (0mg/dL glucose) and maximal range of their sensor, to better highlight the efficacy of this system to medical professionals and as a sign of the sensitivity of their machine

Response:

Thank you for the comment. Required calculations and elaboration on the results were given in the revised manuscript as follows:

The lowest concentration of glucose which is reliably detected by the proposed sensor can be expressed by the limit of detection (LOD). The value of LOD defines the lowest detectable quantity of an analyte when the concentration approaches to zero. The response of the sensor in the range of 200-500 mg/dl was found to be well fitted as y = 6.414 - 0.0048x (R2 = 0.972). Consequently, the LOD was determined to be 30.94 mg/dl using LOD = K × S0/S, where K was chosen as 3.3 with respect to 95 % confidence level, S0 is the standard deviation, and S is the slope of the fitting curve.